# Structural Optimization and Temperature Compensation of GMM-FBG Fiber Current Transducer

**Wei-Chao Zhang** [1,2,3,*], **Lin-Heng Li** [1,2] **and Tao Zhang** [3]

1   Key Laboratory of Engineering Dielectrics and Its Application, Ministry of Education, Heilongjiang Provincial Key Laboratory of Dielectric Engineering, Harbin 150080, China; 2320300019@stu.hrbust.edu.cn
2   School of Electrical and Electronic Engineering, Harbin University of Science and Technology, Harbin 150080, China
3   Qingdao HanHe Cable Co., Ltd., Qingdao 266102, China
*   Correspondence: weichao.zhang@hrbust.edu.cn

**Abstract:** In order to improve the sensitivity and accuracy of the giant magnetostrictive material-fiber Bragg gratings' (GMM-FBG) current sensor, in which the magnetostrictive modulator is Terfenol-D, the temperature effects on the FBG center wavelength and GMM magnetostriction coefficient are investigated to initiate an amending scheme in which temperature parameters are introduced into a GMM-FBG sensing model so as to calibrate current values. Based on electromagnetism theory, the magnetic structure is optimized in design to significantly increase the magnetic coupling efficiency and to homogenize magnetic distribution, employing finite element simulations of the electromagnetic field. The relevant experimental platform is constructed with a wavelength demodulation system. At the temperature range of 20~70 °C, response amplitudes of the current sensor are tested under various current values. The experimental results indicate that the sensitivity of the GMM-FBG current sensor decreases with the temperature increment and is also positively correlated to the target current. Through analyzing the response characteristics of the current sensor to temperature variation, a reasonable GMM-FBG sensing amelioration model with a temperature compensation coefficient is established based on a mathematical fitting method, according to which the current detecting accuracy can be increased by 4.8% while measuring 60 A current at the representative working temperature of 40 °C.

**Keywords:** optical fiber sensor; fiber current sensor; temperature characteristics; correction function; sensitivity analysis

## 1. Introduction

The current transducer plays an increasingly important role in the field of electrical measurement and monitoring [1,2]. However, the traditional electromagnetic current sensor has exhibited many serious shortcomings with insurmountable difficulties when it comes to satisfying the needs of modern high-voltage, high-current, and high-power systems [3,4]. Therefore, many studies have been devoted to exploring a new type of current sensor to fulfill the highly developed current power system of online monitoring, high-precision fault diagnosis, and digital networks. Optical current transducer (OCT) technology offers the capability of highly accurate current measurement and online monitoring of power system and has attracted tremendous attention due to its excellent resistance to electromagnetic interference [5,6].

At present, the most advanced optical current transducers reported in the literature are attributed to the Faraday magneto-optic effect mode, Roche coil optoelectronic mixing mode, and optical fiber mode combined with the magnetostrictive effect. The linear birefringence of Faraday magneto-optic current sensor, the unstable precision, and the aging of sensor head caused by temperature are primary issues hindering the practical application of these developed current sensors [7,8]. In practical applications, the current

transducer with a Roche coil in optoelectronic hybrid formalism is still affected by the electromagnetic field of operation environment and the current supply of high-voltage circuits [9]. A magnetostrictive current transducer utilizes the giant magnetostrictive material (GMM) as the sensing unit and loads the magnetic field into the modulated light waves by integrating the fiber Bragg grating (FBG) on the GMM [10,11]. FBG sensors have excellent performance characteristics, such as a simple structure, high sensitivity, and resistance to electromagnetic interference. They are very suitable for some situations with low accuracy. Under the condition of 50Hz power frequency in this article, using FBG sensors is the best choice [12–14]. In the literature, many current sensors based on GMM, such as Terfenol-D, have been presented. Recent studies report the remarkable progresses in the system of demodulation technology, GMM-FBG magnetic circuit structure design, FBG temperature decoupling, hysteresis nonlinear correction, and so on [15–20]. Meanwhile, it is found that the GMM-FBG fiber current transducer is affected by temperature parameters, which is mainly attributed to the temperature sensitivity of FBG, and some decoupling methods are proposed [20]. However, the magnetostrictive coefficient of GMM also varied with temperature. The current measurement sensitivity of a GMM-FBG sensor could be influenced by both FBG wavelength shifting and magnetostrictive coefficient variation. Presently, studies mainly focus on the influence of FBG wavelength shifting, and the combined factors are seldom investigated. To avoid the sensitivity fluctuation, the response amplitude of a current transducer is measured at different temperatures, and the sensitivity characteristics affected by temperature are analyzed to establish an effective mathematical model of temperature sensitivity. Then, the method of sensitivity correction is proposed, and the simultaneous measurement of current and temperature is realized.

## 2. Fundamental Structure and Principle of GMM-FBG Sensor

The GMM-FBG sensor probe exploits annular silicon steel sheets to construct a magnetic accumulation loop, which converges the ring magnetic field formed by the current in the cable. The GMM modulator is Terfenol-D. The annular silicon steel sheet is partially cut off to build the arrangement area for a GMM-FBG sensitive unit, with the non-ferromagnetic support material being placed in the gap region of it. The GMM with epoxy resin-pasted FBG is attached on the top of support material and the tail fiber is guided to the outside of silicon steel sheet to connect the demodulation equipment. The silicon steel sheet sensitive material is equipped with permanent magnets in axial symmetry to provide a DC-biased magnetic field for the current sensor; the schematic structure of the GMM-FBG sensor is shown in Figure 1.

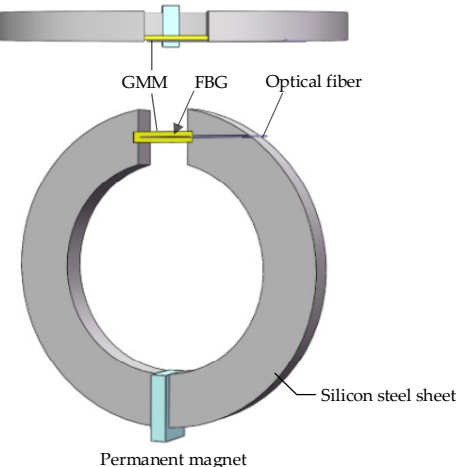

**Figure 1.** Schematic configuration of GMM-FBG current sensor.

GMM is a ferromagnetic material with giant magnetostrictive properties, which possesses the hysteresis characteristics of all ferromagnetic materials. The hysteresis character-

istic curve is shown in the following Figure 2. Silicon steel is an alloy material composed of silicon and steel. The key physical characteristic parameters of GMM and silicon steel sheets are provided here as reference, and the specific parameters are shown in Tables 1 and 2. We assume that the relative magnetic permeability of air is 1.

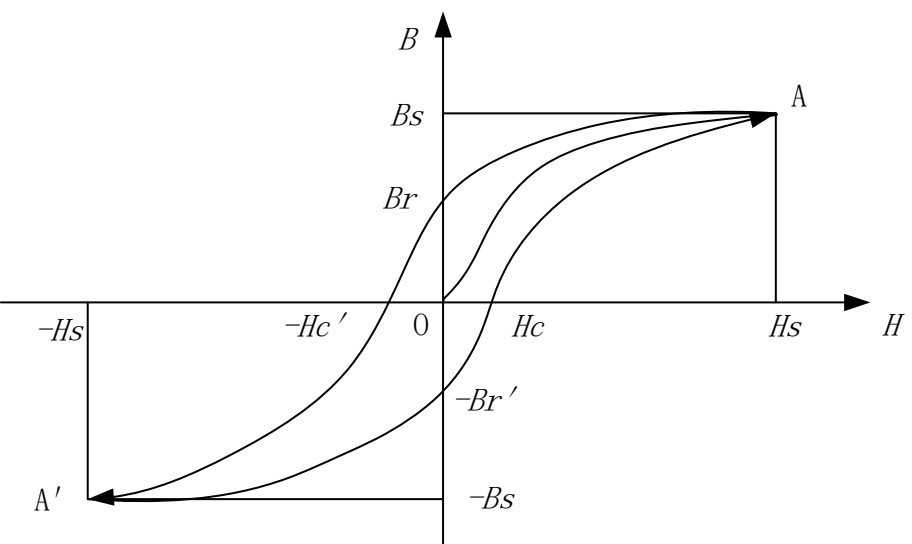

**Figure 2.** The diagram of the hysteresis line characteristics.

**Table 1.** Physical properties of GMM.

| Property | Value |
|---|---|
| Magnetostrictive coefficient (1/m) | $2 \times 10^{-3}$ |
| Toughness index | 2.5 |
| Relative permeability | 8 |
| Saturated magnetic field strength (kA/m) | 200 |

**Table 2.** Physical properties of silicon steel sheets.

| Property | Value |
|---|---|
| Relative permeability | $2 \times 10^{3}$ |
| Saturation magnetic induction (T) | 1.8 |
| Resistivity (μΩ·m) | 100 |
| Magnetostrictive coefficient (1/m) | $2 \times 10^{-8}$ |

FBG can be regarded as a wavelength wideband reflector formed in the fiber core, which reflects part of the light wave incident to the grating. The central wavelength of the reflection can be expressed as follows:

$$\lambda_{\mathrm{B}} = 2n_{eff}\Lambda \tag{1}$$

where $\Lambda$ denotes the period of the optical grating and $n_{eff}$ symbolizes the effective refractive index of the FBG waveguide, both of which are functions of strain and temperature. When temperature is not taken into account, the relationship between the axial strain $\varepsilon$ of FBG and the central wavelength drift $\Delta\lambda_B$ is as follows:

$$\Delta\lambda_{\mathrm{B}} = \lambda_{\mathrm{B}}(1 - P_e)\varepsilon \tag{2}$$

where $P_e$ indicates the effective elastic-optical coefficient, with a value of 0.22 for the silicon fiber medium. According to the principle of electromagnetic induction, an alternating

annular magnetic field will be formed around the cable center when the cable placed inside the closed ring of the GMM-FBG sensor is supplied with AC current $i(t)$. The magnetic field intensity $H$ is determined as follows:

$$H = \frac{i(t)}{2\pi r} \tag{3}$$

where $r$ is radius of the silicon steel sheet. Because the magnetic field assembling structure of the sensor is designed with notched magnetic extraction and the magnetic permeability of GMM is lower than that of silicon steel, the coupling magnetic field in the sensory unit of the GMM sensor will be lower than the magnetic field intensity $H$ due to magnetic leakage described by the coefficient symbolized as $\beta$. Thus, the actual magnetic field intensity of GMM coupling is $\beta H$. Since the sensor is only dominant in a small region of the GMM magnetostrictive strain curve, the axial strain driven by the magnetic field $\beta H$ can be expressed as follows:

$$\varepsilon = K_1 \beta \frac{i(t)}{2\pi r} \tag{4}$$

where $K_1$ identifies the magnetostrictive coefficient of GMM. Without the consideration of temperature's influence, the relationship between the measured current $i(t)$ and the central wavelength of FBG can be written as follows:

$$\Delta \lambda_{\text{B}} = \frac{K_1 \beta \lambda_{\text{B}} (1 - P_e) i(t)}{2\pi r} \tag{5}$$

Accordingly, the variation of the FBG center wavelength is approximately proportional to the measured current. Therefore, the cable current can be obtained by demodulating the frequency and amplitude of the FBG central wavelength.

### 3. Simulation and Design of Magnetic Circuit System

The magnetic field of the closed coil can be calculated accurately by analytic function, although it is difficult to precisely calculate magnetic field from the notched structure of the silicon steel sheet. In order to improve the sensitivity of the sensor and the magnetic accumulation effect of magnetic circuit, the structure of magnetic circuit is optimized with the electromagnetic field finite element analysis method to increase the density of magnetic flux through the GMM and the uniformity of magnetic field inside the GMM. Using the simulation software COMSOL for multiple physical field coupling, the sensor model is constructed, consisting of the circular magnetic loop with 112 mm inner and 72 mm outer diameters in 20 mm thickness and the GMM with a size of 18 mm × 3 mm × 3 mm, for which the relative permeabilities are set as 2000 and 5–10, respectively, under the bus current of 50 A.

The calculated results of magnetic flux density distributions of three magnetic circuit structures with the variation curves showing the magnetic field inside the GMM are shown in Figures 3–5, respectively. When the plane structure is adopted for the notch of the silicon steel sheet, the magnetic flux density through the center region of GMM is about 0.025 T (relatively uniform) while the flux density of GMM through the other terminal is decreased due to the air gap of 0.5 mm to the right ferromagnetic loop, resulting in asymmetric distribution of the magnetic flux density passing through GMM. as illustrated in Figure 3.

When the four-prism structure is used to connect the silicon steel sheets on both sides of the GMM, the flux density distribution through the GMM center shows appreciable inhomogeneity, as shown in Figure 4. The magnetic flux density increases and decreases to 0.0288 T and 0.022 T at one terminal and in the middle segment of GMM, respectively, and the gap between the GMM right terminal and the ferromagnetic loop causes the flux density passing through the GMM to decrease to 0.023 T. However, the magnetic flux density of the GMM terminal near the four-prism structure increases to 0.031 T when the magnetic circuit of silicon steel sheet on one side of GMM is designed as a four-prism structure, as shown in Figure 5. The asymmetric structure of the magnetic circuit and the gap between the GMM

and silicon sheet circuit lead to a rapid decrease in magnetic flux density through the GMM center from 0.031 T at the left terminal to 0.021 T at the right terminal.

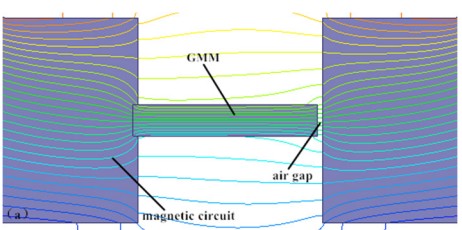
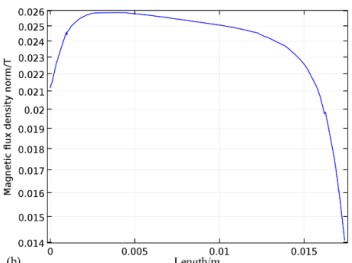

**Figure 3.** Double large section structure: (**a**) magnetic flux density distribution; (**b**) magnetic flux density varying curve along the central axis of the GMM.

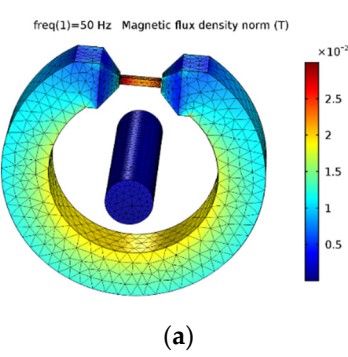
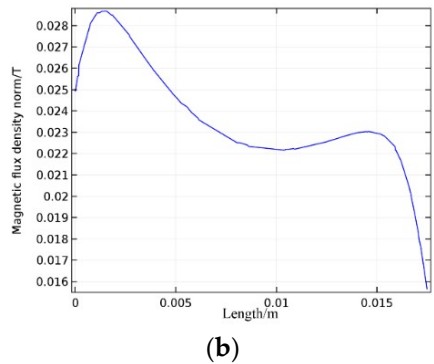

(**a**) (**b**)

**Figure 4.** Double−sided four−prism structure: (**a**) magnetic flux density distribution; (**b**) magnetic flux density varying curve along the central axis of the GMM.

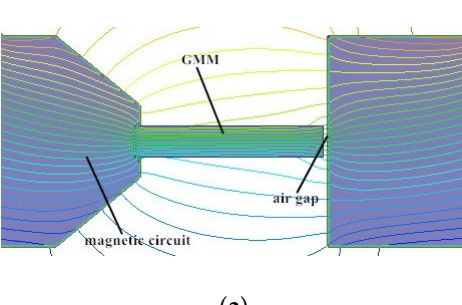
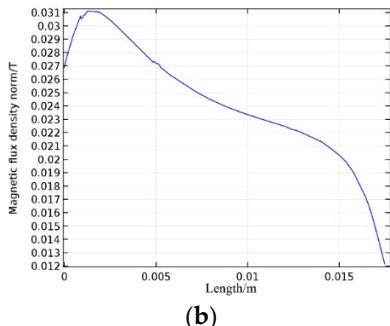

(**a**) (**b**)

**Figure 5.** Asymmetric structure of the four-sided and large section: (**a**) magnetic flux density distribution; (**b**) magnetic flux density varying curve along the central axis of the GMM.

Based on simulation analysis, the air gap between the GMM and silicon steel sheet will cause considerable magnetic flux leakage and lead to the abatement of the coupling magnetic field in the GMM. Furthermore, the small double-end contact surface of the GMM will also reduce the magnetic field in the middle segment of the GMM. Consequently, the magnetic accumulation structure of the current transducer is specially designed as shown in Figure 6, in which a four-prism structure is adopted for the GMM to collect the magnetic field. The cylindrical ferrite is equipped as the guide pole to connect the ferromagnetic loop on the other side. The magnetic field is collected in the GMM, with more uniform flux density passing through the GMM center, as shown in Figure 6b. It is indicated that the flux density through the middle segment of the GMM increases significantly to 0.029 T. Meanwhile, the flux density of the GMM on the side near air gap of the magnetic circuit is enhanced to 0.03 T. Therefore, this designed structure will evidently increase and homogenize the magnetic flux density passing through the GMM.

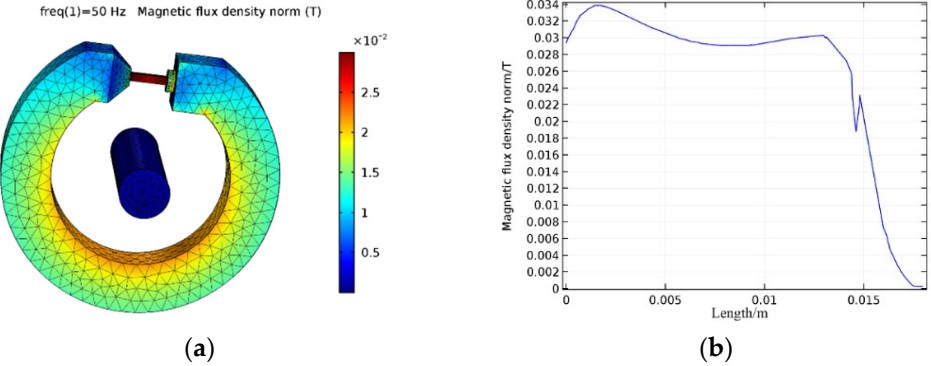

**Figure 6.** Optimized structure: (**a**) magnetic flux density distribution; (**b**) magnetic flux density varying curve along the central axis of the GMM.

In the above results, Figures 3b and 6b are similar. To compare the flux density of the GMM, the root mean square (RMS) method is used to measure the degree of deviation of the data. By collecting the same ten points in Figures 3b and 6b and calculating their respective RMS values, Table 3 can obtain a more uniform result for the latter. Equation (6) is as follows:

$$S^2 = \frac{1}{n}\sqrt{\sum_{i=1}^{10} x_i{}^2} \tag{6}$$

Use $x_i$ ($i = 1,2, \ldots, 10$) to represent each sampling value point, and $S^2$ to represent the RMS value. The $S^2$ parameter for the optimized structure shown in Figure 6b is smaller than the double large section structure shown in Figure 3b. The simulation results indicate that the magnetic flux density of the GMM in Figure 6b shows a low dispersion. The optimized structure designed in Figure 6 homogenizes the magnetic flux density in the GMM.

**Table 3.** RMS of the magnetic flux density along the central axis of the GMM.

| | Double Large Section Structure | Optimized Structure |
|---|---|---|
| $x_1$ | 0.249 | 0.0330 |
| $x_2$ | 0.258 | 0.0332 |
| $x_3$ | 0.259 | 0.0305 |
| $x_4$ | 0.258 | 0.0290 |
| $x_5$ | 0.254 | 0.0288 |
| $x_6$ | 0.250 | 0.0290 |
| $x_7$ | 0.249 | 0.0300 |
| $x_8$ | 0.244 | 0.0301 |
| $x_9$ | 0.237 | 0.0282 |
| $x_{10}$ | 0.226 | 0.0210 |
| $S^2$ | 0.0786 | 0.0093 |

## 4. Temperature Characteristics and Analysis

### 4.1. GMM-FBG Sensor Experiment System

The FBG of the sensor presents a central wavelength of 1546.54 nm with a grating range of 8 mm length, as in Figure 7a, which shows the internal structure of the current sensor. The GMM is designed in a cuboid shape with a size of 20 mm × 3 mm × 2 mm, which is pasted to the FBG by epoxy. The 20 mm packed silicon steel sheets with inner and outer diameters of 650 mm and 900 mm, respectively, are utilized to induce magnetism. The naked GMM-FBG is loaded into a glass tube so as to avoid pollutants coming from the pouring sealant outside when it is being encapsulated in the sensor box, as illustrated in Figure 7. Figure 7b presents the image of a packaged sensor. In order to assemble sensors at an industrial site, the sensor is cut into two symmetrical parts.

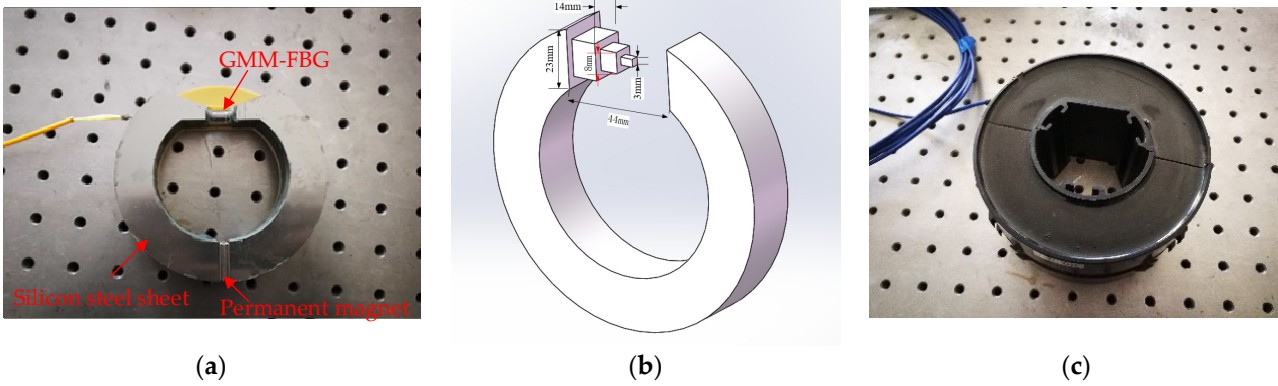

(**a**)  (**b**)  (**c**)

**Figure 7.** Images of sensor: (**a**) internal structure diagram; (**b**) internal guiding structure; (**c**) packaged sensor with epoxy.

The GMM-FBG fiber optic current transducer is composed of two parts: a sensor probe and a photoelectric measurement system, as schematically shown in Figure 8. The sensing probe exploits the sensor structure mentioned above. The photoelectric measurement system consists of tunable fiber laser (DFB-1550 made by Innolume), magnetic coupler, photodetector (Thorlabs PDA 10CS-EC, manufactured by Thorlabs Inc. in Newton, New Jersey, United States), wave filter, data acquisition system, demodulation algorithm module, and controlling PC computer. The fiber laser has a 3 MHz linewidth, and the sweeping range and sweeping rate are 3 nm and 1 kHz, respectively. The acquisition card uses USB-4431 (NI Corp, Austin, Texas, USA). The constructed system operates with a rated test current of 100 A, in which a step-down transformer of 2 kW power is used to generate a larger current; the power source of the alternating current (220 V) is connected to a voltage regulator to produce the voltage from 0 V to 250 V, as in the physical photograph exhibited in Figure 9. The output of the voltage regulator is short-circuited to the input terminal of the step-down transformer. A cable is connected between two sides of the output terminal. The current generated in the cable can be controlled by tuning the voltage regulator. For analyzing response characteristics of the sensor at different temperatures, the current sensor is placed in a thermostatically controlled chamber.

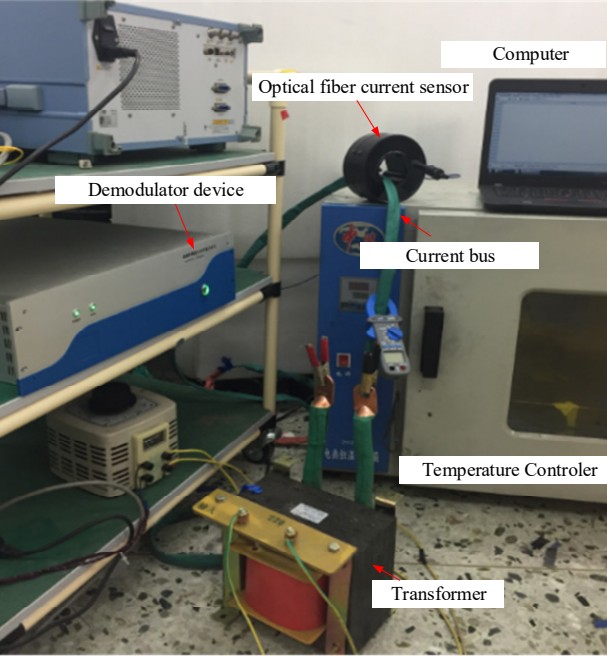

**Figure 8.** Photo of the measurement system.

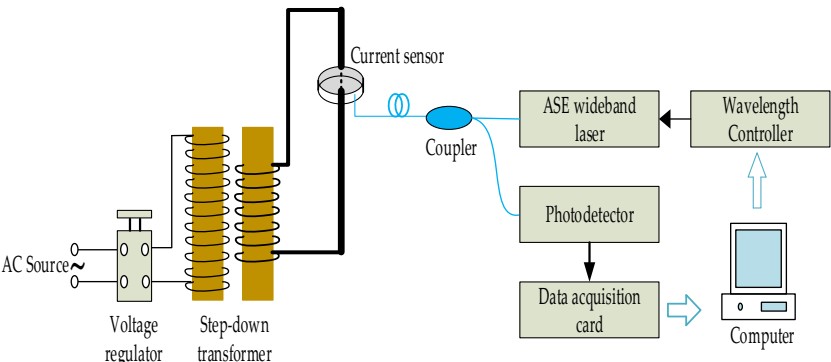

**Figure 9.** Schematic measurement system of the current sensor.

### 4.2. Packaged FBG Temperature Sensitivity Experiment

Although the theoretical temperature drift of the FBG center wavelength is about 10 pm/°C, the actual shift quantity will change in virtue of the packaging process when the GMM-FBG sensor is encapsulated in epoxy resin. In order to delve into the detailed relationship between the central wavelength of the FBG and temperature after epoxy encapsulation, the FBG was pasted on the surface of the GMM at room temperature (20 ± 1 °C), and then tested in the thermostatic control chamber after the epoxy was completely cured. A spectrograph (AQ6370D) was used to test the FBG spectra. The temperature of the thermostatic box was controlled so that it increased from 20 °C to 70 °C with interval steps of 5 °C. The peak value of the FBG spectra was 1546.54 nm at the initial time, while the central wavelength of the FBG increased up to 1547.58 nm with the increasing temperature, as indicated by the shifting spectra of the FBG at different temperatures, as shown in Figure 10.

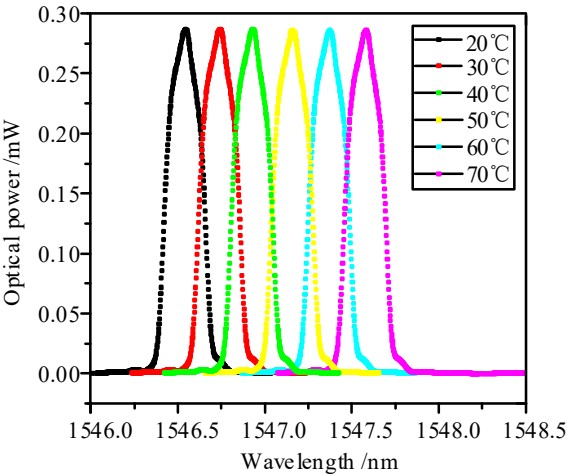

**Figure 10.** FBG spectra at different temperatures.

The tested results as plotted for the FBG central wavelength versus temperature are presented in Figure 11. It is explicitly implied that the FBG central wavelength varies linearly with temperature, according to which the mathematical function of the GMM-FBG wavelength to temperature is achieved by linear fitting as follows:

$$\lambda_0 = 0.02114T + 1546.1 \tag{7}$$

where $\lambda_0$ represents the central wavelength of FBG and $T$ signifies the ambient temperature. The temperature-induced shift in center wavelength reaches 21.14 pm/°C, with the linear correlation coefficient of fitting function approaching 0.9989. The temperature sensitivity

of the FBG center wavelength is higher than the theoretical value, which is primarily attributed to the thermal expansion of the GMM and the cured epoxy resin adhesive.

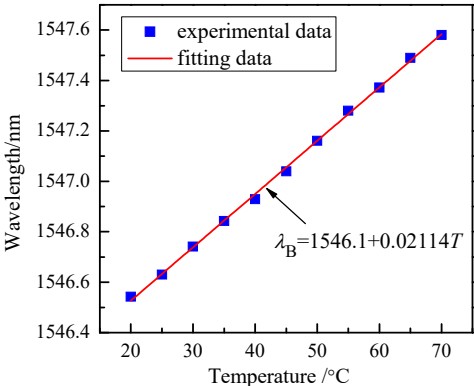

**Figure 11.** FBG temperature characteristic curve after packaging.

*4.3. Current Response Comparison at Different Temperatures*

According to the operation principle of GMM-FBG, the AC current will drive the system to output the optical signal appearing as sine varying function around the FBG central wavelength with the same frequency, and the variation amplitude of the output wavelength is positively correlated with current value. When the ambient temperature is 20 °C, the current meter with 0.5% precision is selected as the standard instrument of current measurement. By adjusting the voltage regulator to output the current of 0~100 A with 5 A steps, the output of the optical fiber demodulator is recorded to obtain the amplitude of sinusoidal variation. Then, the measured current 100 A is gradually diminished to 0 A, and the sinusoidal amplitude of light wavelength is recorded at the same time. In order to reduce the experimental error, the reciprocating test of positive and reverse course is carried out many times to take averaged value of multiple measurements. The sensor probe is placed in a constant temperature box to perform the current detection experiments at temperatures of 30 °C, 40 °C, 50 °C, and 60 °C, respectively.

The changing wavelength of the FBG in the sensor includes sinusoidal signals and bias signals, which are modified by the tested current and temperature, respectively. Based on the filter principle, the changing wavelength signals are integrated and divided by integrating time, so as to obtain the final result of the bias signal, which is a function of temperature and independent of current, as shown in Figure 12. Eventually, the temperature values can be achieved from the wavelength–temperature relationship provided by the curves in Figure 11. Therefore, the sinusoidal signals (amplitude and periodicity of changing wavelength) that are determined by the current will not affect the temperature measurement.

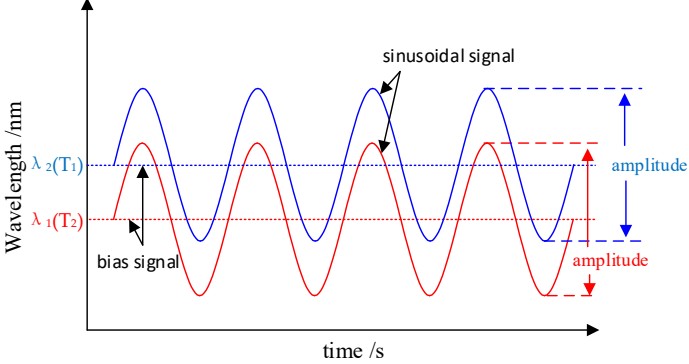

**Figure 12.** The sinusoidal signals and bias signals rendered in the changing wavelength of the FBG sensor for different temperatures.

The experimental results of the relationship between the current and the sinusoidal amplitude of changing wavelength for FBG at different temperatures are illustrated in Figure 13. As the temperature increases, the slope of transfer function of the GMM-FBG fiber current sensor and the corresponding dominant current are both reduced, resulting in faster debasement of sensitivity.

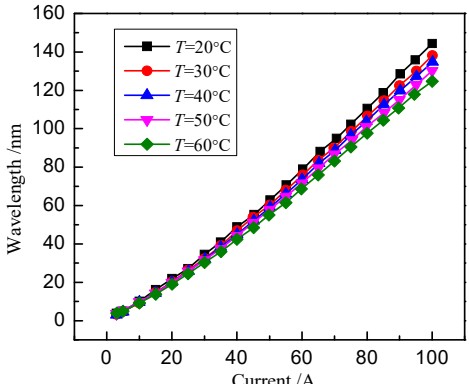

**Figure 13.** Amplitudes of changing wavelength versus current at different temperatures.

As shown in Figure 14, the relation between the variation amplitude of light wavelength and temperature demonstrates that the sensor sensitivity deteriorates with the increase in temperature under constant current. The varying amplitude of sinusoidal light wavelength decreases by 3.01% and 2.67% for every incremental 10 °C under the measured currents of 40 A and 60 A, respectively. Moreover, the sensitivity decreases faster with increasing temperature for higher measured currents. The saturation magnetostriction of GMM also decreases linearly with the increase in temperature, so that the output response of the GMM-FBG sensor under the same driving current declines and consequently reduces in sensitivity.

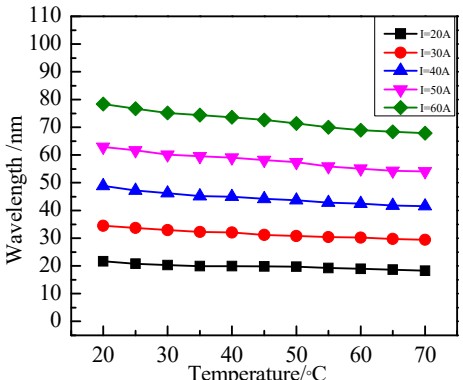

**Figure 14.** Amplitudes of changing wavelength vary with temperature under various currents.

### 4.4. Current Calibration Scheme

The current response analysis makes it clear that the temperature change will cause the output signal of GMM-FBG sensor to change in two aspects: one is the drift of the FBG central wavelength, while the other is the change in the wavelength variation amplitude under the same current. In the detection of alternating power current at the 50 Hz frequency, the central wavelength of FBG only provides the DC component, the change in which does not affect the demodulation of the AC current. According to the measured relationship between the FBG central wavelength and ambient temperature as depicted in Figure 15, the sensor temperature can be tested by demodulating the central wavelength of the DC component in response to the light wavelength. The change in light wavelength varying amplitude caused by shifting temperature affects the accuracy of the current measurement.

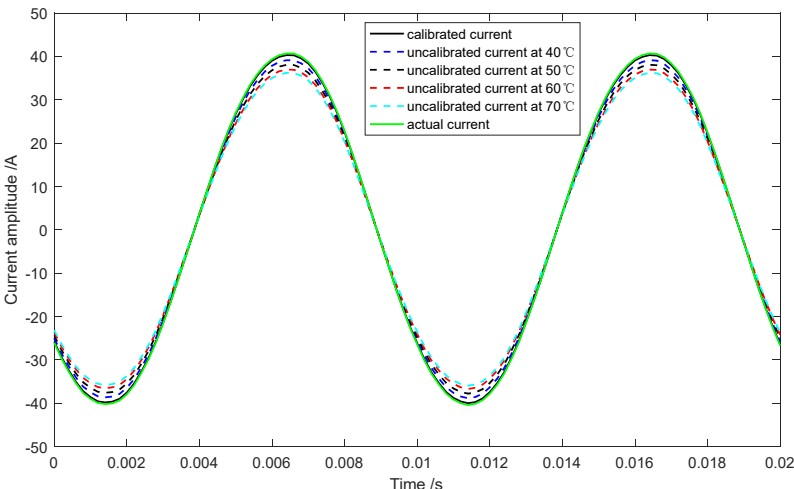

**Figure 15.** Measured current (calibrated current in comparison with non−calibrated current).

The optical wave amplitude of the current response is a two-parameter function of temperature and current, motivating us to supplement temperature calibration parameters to the traditional wavelength amplitude as a function of current, as in the following formula:

$$P = \text{F}(T, i(t)) \tag{8}$$

where $P$ signifies the sinusoidal amplitude of light wavelength variation, $T$ is the ambient temperature, and $i(t)$ denotes the measured current. Since the ambient temperature $T$ can be obtained by demodulating the DC component of light wavelength (FBG center wavelength $\lambda_0$), the transfer function of GMM-FBG sensor with temperature calibration parameters can be represented as follows:

$$P = \text{F}(T(\lambda_0), i(t)) \tag{9}$$

As the relationship between FBG central wavelength and measured current can be described as a quadratic function, the temperature calibration coefficient can be expressed as follows:

$$P = a(T) \cdot i^2(t) + b(T) \cdot i(t) + c(T) \tag{10}$$

The wavelength response amplitudes of the GMM-FBG sensor driven by 20−60 A current at 30 °C are listed in Table 4. By using quadratic function fitting, the relationship function between wavelength varying amplitude and measured current at this temperature is obtained as follows:

$$P = 0.003423i^2(t) + 1.0710i(t) - 1.6540 \tag{11}$$

Similarly, the wavelength amplitudes of GMM-FBG at the temperature range of 30–60 °C, as a related function of measured current are obtained by functional fitting, and the results are listed in Table 5. According to the relation and analyzing varying trend of item factors in these functions with the temperature, the temperature function of these factors can be obtained by fitting as follows:

$$\begin{aligned} a(T) &= -0.00002T + 0.00412 \\ b(T) &= -0.00289T + 1.1543 \\ c(T) &= -0.00143T^2 - 0.10325T + 0.447 \end{aligned} \tag{12}$$

The GMM-FBG wave amplitudes as a binary function of current and temperature after introducing calibration parameters can be obtained by combining Equation (12) and Formula (10). After obtaining the temperature calibration function of current sensor, the measured current is adjusted to 40 A and the ambient temperature is controlled at 40 °C for the experimental system. The output current of the GMM-FBG system using the

temperature calibration function is 40.29 A with a minimal deviation of 0.98%, in substantial contrast to the much larger deviation of 3.80% for the measured current value of 39.12 A without the calibration of temperature compensation. In addition, in order to increase the reliability of the experiment, multiple sets of temperature compensation experiments have been performed. The comparison curves of temperature compensation at different temperatures are depicted in Figure 15. As can be seen from the data in the figure, the errors between the uncompensated current and the actual current at different temperatures (50 °C, 60 °C, and 70 °C) are 6.49%, 9.18%, and 11.04%, respectively. It can be seen that compared to the calibrated values, the error between the uncalibrated values and the true values is relatively large. In addition, we can also observe a pattern from the data graph that, as the temperature increases, the current amplitude shows a decreasing trend.

Meanwhile, the frequency spectrum of the measured current is calculated using the fast Fourier transform method. The peak value occurs at the frequency of 50 Hz, which is identical to the tested current, without any other modes involved in the measurement system, as in the results shown in Figure 16.

**Table 4.** The transfer function of the current sensor at 30 °C.

| Physical Quantity | Values | | | | |
|---|---|---|---|---|---|
| Temperature/°C | | | 30 | | |
| Current/A | 20 | 30 | 40 | 50 | 60 |
| Amplitude/pm | 20.586 | 32.952 | 46.372 | 60.485 | 75.429 |

**Table 5.** The transfer function of the current sensor at various temperatures.

| Temperature/°C | Fitted Function |
|---|---|
| 20 | $P = 0.003806i^2(t) + 1.085i(t) - 0.9263$ |
| 30 | $P = 0.003437i^2(t) + 1.076i(t) - 1.6030$ |
| 40 | $P = 0.003317i^2(t) + 1.042i(t) - 1.4080$ |
| 50 | $P = 0.003093i^2(t) + 1.024i(t) - 0.8944$ |
| 60 | $P = 0.002991i^2(t) + 0.967i(t) - 0.7307$ |

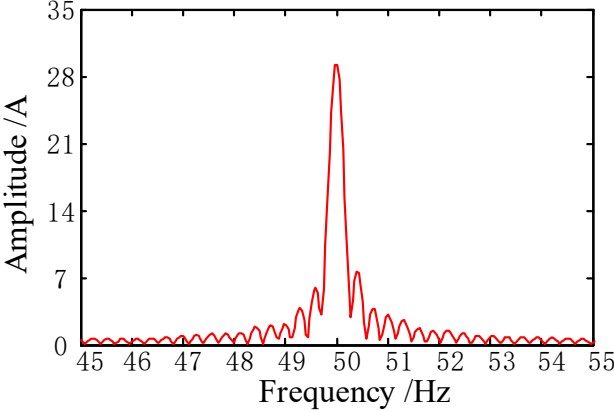

**Figure 16.** Frequency spectrum of the measured current.

## 5. Conclusions

Based on the optimal design of a magnetic circuit of a GMM-FBG optical fiber current sensor, the effects of the magnetostrictive coefficient varying with temperature on the sensitivity and accuracy of current detection are emphatically analyzed. A magnetically coupled structure with high magnetic concentration and uniform distribution in the GMM sensing region is designed by electromagnetic field simulations with the finite element analysis method. The temperature experiments of the GMM-FBG sensor demonstrate that the sensor sensitivity decreases with the increase in temperature, which is positively

correlated with the measured current value. By analyzing the temperature characteristics, a mathematical model of the current and output wavelength amplitude of the GMM-FBG current sensor with temperature calibration parameters is established. The current measurement results at 40 °C verify that the accuracy of current measurement has been significantly improved by the proposed amelioration model.

**Author Contributions:** Conceptualization, W.-C.Z.; methodology, W.-C.Z.; software, L.-H.L.; validation, T.Z. and L.-H.L.; formal analysis, W.-C.Z.; investigation, W.-C.Z.; resources, W.-C.Z.; data curation, W.-C.Z.; writing—original draft preparation, L.-H.L. and T.Z.; writing—review and editing, W.-C.Z.; supervision, W.-C.Z.; project administration, W.-C.Z.; funding acquisition, W.-C.Z. All authors have read and agreed to the published version of the manuscript.

**Funding:** Grand No. KFKT202210 from Key Laboratory of Special Machine and High Voltage Apparatus (Shenyang University of Technology), Ministry of Education.

**Institutional Review Board Statement:** Not applicable.

**Informed Consent Statement:** Not applicable.

**Data Availability Statement:** No additional data are available.

**Conflicts of Interest:** Author Tao Zhang was employed by the company Qing Dao HanHe Cable Co., Ltd. The remaining authors declare that the research was conducted in the absence of any commercial or financial re-lationships that could be construed as a potential conflict of interest.

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
