# Peer review of "Structural Optimization and Temperature Compensation of GMM-FBG Fiber Current Transducer"

_photonics, doi:10.3390/photonics10121376_

Round 1

Reviewer 1 Report

Comments and Suggestions for Authors

The authors presented a corrected manuscript entitled “Structural Optimization and Temperature Compensation of GMM-FBG Fiber Current Transducer”. The research work is interesting and could be published in Photonics.

Overall, this paper is well-written with an organized structure, clear explanation, copious graphs, and sufficient data. But several suggestions are supplied:

1. Silicon steel sheets were used as magnetic conductive materials in the study. Can other materials replace silicon steel?

2. In the paper, a four-prism structure is adopted for to collect the magnetic field. Does the author consider any other structures gather the field?

3. In Fig.6., the “Schematic photo” can be “photos”.

4.  In section 4.4, “make” should be “makes” in the first sentence.

5 .  In Table 2, the corresponding content of two columns should be aligned.

6.  In Line 294, the format should be unified.

7.  Suggest adding testing equipment model for figure 9.

8. In equation(5), I find that some letters do not provide a physical explanation.

Author Response

1. Silicon steel sheets were used as magnetic conductive materials in the study. Can other materials replace silicon steel?

ReviseSilicon steel sheets are commonly used magnetic material for power transformers and current transformers. Silicon steel sheets have the advantages of high magnetic permeability, easy processing, and low cost. So silicon steel sheets were suitable for optical fiber current transformers in the manuscript.

Actrually, there are also some other materials with high magnetic permeability that can be used for this sensing, such as Ultrafine Crystalline and ferrite. Different materials have different characteristics. Ultrafine Crystalline and ferrite both have high response frequency compare to Silicon steel, but they are more expensive. In China, the frequency of the power system is 50Hz. Silicon steel sheets are sufficient to meet the requirements of current sensors in the manuscript.

2.In the paper, a four-prism structure is adopted for to collect the magnetic field. Does the author consider any other structures gather the field?

ReviseIn order to obtain current transformers with high sensitivity and stable performance, the magnetic conductor structure needs to efficiently couple the magnetic field while ensuring that the magnetic induction lines in the GMM material are relatively uniform. Based on simulation analysis, it was found that the proposed structure has good performance. Of course, similar structures may also exist through attempts at other shapes.

3.In Fig.6., the “Schematic photo” can be “photos”.

ReviseRevised in the manuscript.

4.In section 4.4, “make” should be “makes” in the first sentence.

ReviseThe grammar error in the first sentence of section 4.4 has been corrected.

5. In Table 2, the corresponding content of two columns should be aligned.

ReviseThe content in Table 2 has been aligned as required.

6.In Line 294, the format should be unified.

ReviseThe formatting issue with capitalization in the title of Table 2 has been corrected.

7.Suggest adding testing equipment model for figure 9.

ReviseRevised in the manuscript.

8.In equation(5), I find that some letters do not provide a physical explanation.

ReviseThe letters in the formula are explained in the previous text (in the formula explanation section above).

Reviewer 2 Report

Comments and Suggestions for Authors

This paper mainly reports the influence of temperature on the central wavelength of FBG and the magnetostriction coefficient of GMM. The chart format, simulation and experimental data of this article are well represented. However, I think there are still many problems that need to be modified.

1.      In the simulation experiment in Section 3, a and b in Figure 2 respectively represent the magnetic flux density and the curve of magnetic flux density change along the central axis of GMM. The article's description of Figure 2 is “asymmetric distribution of magnetic flux density passing through GMM”, However, the simulation results of the improved current sensor shown in FIG. 5 are like those shown in FIG. 2, and it cannot be seen whether they are more uniform. It is suggested to increase the comparative data to support the views of the paper.

2.      In this paper, the variation curves of the central axis of magnetic flux density extension GMM under different structures were measured by simulation experiments, to find the structure with the largest magnetic accumulation to increase the sensitivity of the sensor. However, insufficient comparative experiments were conducted to verify the claim that this structure improved the sensitivity in the paper. Therefore, it is suggested to supplement experiments to enhance the credibility of the views in the paper.

3.      Figure 14 shows the comparison between the calibrated measured current and the uncalibrated measured current. There should be three curves, namely: the actual current curve, the calibrated measured current and the uncalibrated measured current.

4.      Another important part of the author's article is the temperature compensation for the measured current. The results of temperature compensation at 40℃ are mentioned in the paper, and the effect is good. Are there any experimental results at other temperature conditions? It is hoped that the measured results after compensation at different temperatures can be shown in graphs.

Reviewer 3 Report

Comments and Suggestions for Authors

The authors have done an interesting work based on simulation and experimental work. On my opinion, the paper can not be accepted if the GMM is not specified in order to consider the complete experiment and its results. In the work May be appropiate comment about the magnetic hysteresis of the silicon steel and of the GMM, and how it can affect to the measurements. Photos on fig.6 and 7 must be improved to help the reader to get extra information about the setup and sensor. The text could be improved in order to help to understand the work done. Could be plotted the relation between current and magnetic field on the GMM. 

Reviewer 4 Report

Comments and Suggestions for Authors

This manuscript aims to enhance the sensitivity and accuracy of a giant magnetostrictive material-fiber Bragg grating (GMM-FBG) current sensor by considering the temperature effects on its components. A magnetically coupled structure is refined for heightened magnetic concentration and uniformity using finite element analysis. Experiments confirm that the sensor's sensitivity to current decreases as temperature increases, yet the sensitivity maintains a positive correlation with the current being measured. A mathematical model incorporating temperature calibration parameters is established to improve the current measurement accuracy at 40ºC. The article presents intriguing conclusions that have practical relevance. However, several technical issues need to be addressed to improve the quality and clarity of the paper. Therefore, I recommend a major revision.

The following are my specific comments and questions for the authors:

1. The English language quality of this article is good but requires further refinement. I recommend that the authors conduct a thorough review to ensure correct grammar and expression.

2. The abbreviation GMM-FBG is used in the Abstract without prior introduction of its full name. Please provide the full term in both the Abstract and the main body of the article before employing the abbreviation.

3. The article's literature review would benefit from a broader introduction to the functionality of FBGs and their significance in practical applications. Considering that numerous optical techniques exist for measuring temperature, what led to the choice of FBGs for the sensor? Incorporating insights from additional sources, such as the ones suggested below, would provide readers with a more well-rounded perspective on the topic. 

[1] Long-term stabilities fiber Bragg grating (FBG) arrays inscribed by femtosecond lasers at 910C. CLEO 2021.

[2] Real-time optical fiber-based distributed temperature monitoring of insulation oil-immersed commercial distribution power transformer. DOI: 10.1109/JSEN.2020.3024943.

[3] High-speed interrogation of embedded fiber Bragg grating (FBG) sensors fabricated by ultrasonic additive manufacturing. OFS 2022.

4. In section 4 on page 5, the make and model of the optical components (laser source, PD, DAQ, etc.) should be specified. What is the linewidth, sweep range, and sweep rate of the tunable laser source? And what is the sampling rate of the DAQ?

5. The FBG’s central wavelength decreases as the temperature goes up in Figure 9.

6. The text in lines 201-205 is repetitive.

7. In practical applications, the central wavelength shifts of FBGs are influenced by both temperature and strain. The article does not consider the effects of strain. I wonder if variable strain during the experiment has affected the sensor's performance. This should be discussed in the paper.

Comments on the Quality of English Language

The English language quality of this article is good but requires further refinement. I recommend that the authors conduct a thorough review to ensure correct grammar and expression.

Round 2

Reviewer 2 Report

Comments and Suggestions for Authors

In this paper, the influence of temperature on the central wavelength of FBG and the magnetostriction coefficient of GMM is discussed. In view of the previous review comments, the author has made a detailed reply, and the experimental part of the article has also been effectively supplemented. I think the article has been revised to meet the standards of the journal, it can be accepted by the photonics journal.

Author Response

Thank you for the evaluation. 

Reviewer 3 Report

Comments and Suggestions for Authors

Dear authors,

for future works my recomendation is to be more rigorous with the measurements and characterization of all the materials implied in the research. In this papers is not clear justified the characterization of tefenol-D which is an important component of your researc. It is not necessary to explain basic theory about magnetism (like the histeresys loop) in the paper but show the complete characterization of materials employed.

Reviewer 4 Report

Comments and Suggestions for Authors

The revised manuscript by the authors has made progress in addressing previous concerns. However, further clarification is needed before accepting the manuscript. My questions are outlined below:

1. In the manuscript, it is stated that the tunable laser's wavelength sweep range is 3 nm. I would like to know the operational temperature range of the GMM-FBG sensor, as well as its maximum strain capacity under practical conditions. Given the coefficient of 21.14 pm/C, a 3 nm range equates to around 150 degrees Celsius. This might be insufficient if the sensor operates at higher temperatures, considering the necessity to accommodate additional strain-induced wavelength shifts.

2. The details of bibliographic references should be checked carefully. This includes verifying the accuracy of authors’ names, publication issues, page numbers, conference locations, and years. Ensuring the integrity and accuracy of these references is crucial for the manuscript's credibility and utility to readers.

Comments on the Quality of English Language

The manuscript's English is generally good, though it contains a few minor errors. These do not detract from the overall readability of the article.
